# ROMO1: A Distinct Mitochondrial Protein with Dual Roles in Dynamics and Function

**DOI:** 10.3390/antiox14050540

**Published:** 2025-04-30

**Authors:** Angel Yordanov, Eva Tsoneva

**Affiliations:** 1Department of Gynaecological Oncology, Medical University Pleven, 5800 Pleven, Bulgaria; 2Department of Reproductive Medicine, Specialized Hospital for Active Treatment of Obstetrics and Gynaecology Dr Shterev, 1330 Sofia, Bulgaria; dretsoneva@gmail.com

**Keywords:** ROMO1, ROS, oxidative stress, cancer, cervical cancer

## Abstract

Reactive oxygen species modulator 1 (ROMO1) is a nuclear-encoded inner mitochondrial protein known for its dual role as a modulator of reactive oxygen species (ROS) and a non-selective ion channel. Initially identified for its role in ROS production, ROMO1 has garnered attention for its functional properties as a non-selective ion channel that regulates ion homeostasis in mitochondria. This article examines ROMO1 from both perspectives, emphasizing its structural and functional characteristics, physiological roles, and implications in health and disease. Understanding ROMO1’s dual functionality provides insight into its potential as a therapeutic target for oxidative stress-related disorders, especially cancer progression.

## 1. Introduction

ROMO1 is a small nuclear-encoded inner mitochondrial protein involved in crucial cellular processes, including redox signaling, mitochondrial dynamics, and ion transport [1]. Other names include hGlyrichin, Epididymis tissue protein Li 175, and Mitochondrial-targeting GxxxG motif protein (MTGM2) [2]. Initially discovered as a key player in ROS generation, subsequent studies revealed its ability to form non-selective ion channels in mitochondrial membranes, suggesting a broader physiological role.

## 2. Structural and Molecular Characteristics

### 2.1. ROMO1 as a Protein

ROMO1 consists of 79 amino acids and is located on chromosome 20q11.22 [3]. Research by Lee et al. in 2018 highlighted various functions of ROMO1, identifying that it has two transmembrane domains (TMDs) characterized by α helices connected by a basic loop [1]. TMD1 is composed of a hydrophobic α helix, while TMD2 contains polar amino acids (K58, T59, Q62, S63, T66, and T69), each separated by other amino acids, with an average of 3.6 amino acids per α-helical turn. Notably, the amino acid sequences of ROMO1 in 82 out of 247 studied animal species closely resemble that of Homo sapiens ROMO1, suggesting a conserved sequence across species. Due to ROMO1’s small size and hydrophobic characteristics, its structure could not be determined through X-ray crystallography or nuclear magnetic resonance (NMR) techniques; however, structural bioinformatics led to the proposal of a hexameric model for ROMO1. It is localized to the mitochondrial inner membrane, where it interacts with other mitochondrial proteins. Structurally, ROMO1 contains two transmembrane domains, facilitating its integration into the lipid bilayer. Post-translational modifications, such as phosphorylation, are thought to regulate its activity, though detailed structural data remain limited.

### 2.2. ROMO1 as a Channel

Lee et al. demonstrated that ROMO1 can form homo-oligomers within the inner mitochondrial membrane [1]. This protein belongs to a family of virus-encoded non-selective ion channels, classified as class II viroporins, characterized by small α-helical transmembrane proteins that contain one amphipathic helical transmembrane domain (TMD). This structural feature is crucial for the formation of homo-oligomeric, non-selective ion channels. Given the structural similarities between ROMO1 and class II viroporins, it has been proposed that ROMO1 may function as a viroporin-like ion channel in eukaryotes. Since ROMO1 exhibits properties akin to viroporins, its homo-oligomeric form is thought to enhance the permeability of the mitochondrial membrane, similar to viroporins. The TMDs in ROMO1, particularly the C-terminal amphipathic helix TMD2, are essential for its pore-forming activity. As a result, ROMO1 can be inhibited by viroporin inhibitors like hexamethylene amiloride (HMA), but not by amantadine, rimantadine, or N-nonyldeoxynojirimycin. Notably, using HMA as an inhibitor increases the membrane potential (ΔΨm) by blocking ROMO1. Moreover, it has been shown that ROMO1 can function as a non-selective cation channel similar to class II viroporins. Transition metal ions, such as Cu^2+^, Zn^2+^, and Fe^2+^, play significant roles in mitochondrial function, including regulating ROS production. Lee et al. found that Fe^2+^ in the mitochondrial membrane can modulate and reduce ROMO1 activity [1]. Overall, they concluded that ROMO1 may operate as a hexameric, non-selective cation channel, paralleling the function of viroporins (Figure 1).

## 3. Physiological Roles

### 3.1. ROS Production and Regulation

ROMO1 contributes to mitochondrial ROS generation, likely through complex III of the electron transport chain. Notably, a specific basal level of ROS is necessary for normal cellular signaling and proliferation [2]. However, various stressors such as toxins or radiation can further increase ROS production, leading to oxidative damage to proteins, lipids, and DNA [3]. Chronically elevated ROS disrupt redox homeostasis and have been implicated in aging, inflammation, and malignant transformation [3]. While ROMO1-derived ROS are critical for physiological processes, unchecked overproduction of ROS due to dysregulated ROMO1 activity may result in cellular injury. Indeed, studies have linked ROMO1-mediated ROS to cell death under certain conditions. For example, ROMO1 serves as a mitochondrial mediator of tumor necrosis factor-alpha (TNF-α) signaling, producing ROS that trigger apoptosis in the absence of growth factors [4]. Similarly, in lung epithelial cells, elevated ROMO1 expression contributes to oxidative stress-induced cell death, such as that caused by cigarette smoke exposure [5].

### 3.2. Mitochondrial Dynamics

Mitochondria create a dynamic network that is shaped by continuous cycles of fusion and fission among individual organelles, a process referred to as mitochondrial dynamics [6]. Norton et al. performed a study which concluded that ROMO1 plays a pivotal role as a redox-dependent regulator of mitochondrial dynamics [6]. The researchers performed a genome-wide RNA interference (RNAi) imaging screen in HeLa cells to identify regulators of mitochondrial morphology. Using small interfering RNAs (siRNAs) targeting known mitochondrial dynamics proteins (DRP1, MFN1, and MFN2), they were able to induce mitochondrial elongation or fragmentation. Among the key findings, the screen identified several regulators of mitochondrial dynamics, including DRP1, OPA1, MFF, and YMEL1. Notably, three out of four siRNAs targeting the reactive oxygen species modulator ROMO1 led to mitochondrial fragmentation.

To confirm the specificity of the results, the researchers restored normal mitochondrial morphology in ROMO1-knockdown cells by introducing a ROMO1 complementary DNA (cDNA) that was resistant to siRNA knockdown. A mutant ROMO1 (ROMO1-FFAA) with two phenylalanine-to-alanine substitutions failed to rescue mitochondrial morphology, suggesting the importance of specific residues in ROMO1’s function.

Additionally, mitochondria from ROMO1 knockdown cells showed a 50% lower fusion rate compared to controls, indicating that ROMO1 is required for proper mitochondrial fusion. Despite previous findings suggesting that ROMO1 overexpression leads to fragmentation, the researchers found that C-terminally tagged ROMO1 constructs induced fragmentation, suggesting a dominant negative effect.

Norton et al. concluded also that ROMO1 affects the formation of mitochondrial cristae, which are necessary for optimal electron transport chain function.

## 4. Pathophysiological Implications

### 4.1. Cancer

ROMO1 is frequently overexpressed in cancer cells. There are several studies that have reported ROMO1 expression in different cancer cells: colorectal cancer, non-small cell lung cancer, hepatocellular carcinoma, and cervical cancer. All of those studies have tested the expression of this biomarker using immunohistochemistry, except for hepatocellular carcinoma, where the researchers used a combination of methods: flow cytometry, RNA interference, and gelatin zymography [7]. A study conducted by Kim et al. [8] demonstrated that ROMO1 could serve as a prognostic marker for colorectal cancer (CRC). Elevated ROMO1 expression in cancerous tissues was strongly linked to early recurrence and reduced survival rates in patients who underwent curative resection. Furthermore, when stage IV CRC patients were included, ROMO1 expression showed a significant correlation with overall survival. The researchers also found that ROMO1 overexpression was closely associated with a higher lymph node ratio, increased lymphatic invasion of primary tumors, and lower survival rates in CRC patients. The results in patients with lung cancer, where the expression of ROMO1 was also tested using immunohistochemistry, are similar. A study conducted by Kim et al. [9] in 2020, where 98 patients were enrolled, concluded that ROMO1 levels rise considerably as the disease progresses to more advanced stages [9]. Additionally, elevated expression of this protein has been linked to lymph node metastasis, disease progression, and unfavorable prognosis overall. A study conducted in 2012, where 95 patients were enrolled, explored the link between ROMO1 expression, mitochondrial ROS production, and cancer cell invasiveness, particularly in hepatocellular carcinoma (HCC) [10]. Since mitochondrial ROS are associated with tumor invasion, the researchers investigated whether ROMO1, a known inducer of mitochondrial ROS, plays a role in HCC cell invasiveness. [10] To assess this, they used ROMO1 small interfering RNA (siRNA) to knock down its expression in HCC cells (Huh-7 and Hep3B) and MCF-7 breast cancer cells. Flow cytometry confirmed high transfection efficiencies (over 93% in all cell lines). As expected, ROMO1 knockdown reduced intracellular ROS levels, as shown by flow cytometric analysis using MitoSOX staining. Next, the study evaluated whether reduced ROMO1 expression impacted tumor cell invasion. A Boyden chamber invasion assay revealed that ROMO1 knockdown significantly decreased the invasive activity of HCC cells. This finding was further supported by gelatin zymography, which showed a reduction in MMP-9 activity—an enzyme associated with tumor invasion. To validate these results in an in vivo setting, the researchers used a lung metastatic mouse model. Mice injected with Huh-7 cells transfected with ROMO1 siRNA exhibited significantly fewer metastatic lung nodules compared to controls. Collectively, these findings highlight that ROMO1 promotes HCC cell invasiveness by increasing ROS levels, supporting its potential role as a therapeutic target for limiting cancer metastasis. However, when it comes to cervical carcinoma progression and ROMO1, the results are quite different. We conducted a study in 2020 where 75 patients with CC in different stages were enrolled. We used immunohistochemistry to test the expression of ROMO1. Since there is no established scoring system for ROMO1, we used the H-score, Allred score, and Combined score to analyze our findings. In conclusion, the expression of ROMO1 was significantly higher in the early stages of the disease; as the cancer progresses the expression of the protein decreases. We also established that the H-score produced the most significant *p*-value, making it the preferred method for assessing ROMO1 expression via immunohistochemistry.

### 4.2. Neurodegenerative Diseases

As is well described in the literature, oxidative stress refers to an imbalance in redox homeostasis caused by elevated levels of reactive oxygen species [11]. ROS are metabolic byproducts emerging from different cell types, with mitochondrial metabolism being the primary source [3]. Under normal conditions, oxygen is fully reduced to water in the mitochondria. However, in pathological states, electrons escape from the electron transport chain, leading to ROS generation [12]. Environmental stress can further amplify ROS levels. While low ROS concentrations play a role in regulating cellular signaling pathways such as redox signaling, prolonged exposure to excessive ROS can damage proteins, lipids, and nucleic acids [13]. This increased ROS production contributes to mitochondrial dysfunction, which in turn leads to disrupted mitochondrial integrity—a key factor in the progression of neurodegenerative disorders, such as Alzheimer’s and Parkinson’s disease [13,14]. Norton et al. [6] concluded in their study that ROMO1 plays a pivotal role in maintaining the integrity of the mitochondrial cristae junctions. Through an analysis of the inner mitochondria, Norton et al. found that cells with ROMO1 suppression exhibited mitochondria with either significantly reduced or completely absent cristae. In patients with Parkinson’s disease mitochondrial cristae are often reduced or altered [15]. This connection between ROMO1, cristae structure, and neurodegeneration suggests that ROMO1 might influence neuronal cell viability by maintaining mitochondrial architecture.

### 4.3. Cardiovascular Diseases

In a study published in 2024, the authors concluded that ROMO1 is essential for maintaining spare respiratory capacity (SRC) and that its absence inhibits cardiac hypertrophic growth [16]. Additionally, they show that ROMO1 deficiency leads to reduced ROS production and impaired redox-sensitive NF-κB signaling, which in turn prevents cardiomyocyte hypertrophy. Furthermore, while ROMO1 levels increase during the early stages of adaptive hypertrophy, they eventually decline in failing hearts, potentially contributing to SRC loss and the progression of heart failure (HF). Overall, this study offers new insights into HF pathogenesis and may pave the way for novel preventive and therapeutic approaches focused on targeting ROMO1 and restoring SRC [16].

## 5. Conclusions

We believe that ROMO1 could be used as a potential biomarker in many medical fields, especially in cancer progression. Its dual role as a modulator of reactive oxygen species and a non-selective ion channel highlights its importance in maintaining mitochondrial homeostasis and dynamics through ROS production regulation and maintaining the right structure of the mitochondrial cristae. Recent evidence also indicates that ROMO1 is essential for mitochondrial function during early embryonic development, underscoring its fundamental role in cell physiology [17]. Furthermore, given ROMO1’s capacity to transport Fe^2+^, there may be a connection to ferroptosis—an iron-dependent form of cell death characterized by lipid peroxidation [18]. While direct evidence linking ROMO1 to ferroptosis is currently lacking, this possibility merits investigation. Overall, further understanding the structural characteristics and physiological roles of ROMO1 can provide valuable insights into its potential as a therapeutic target for oxidative stress-related disorders. Notably, its role in mitochondrial ROS generation was established early on [19,20], and its evolutionary conservation across species further underscores its fundamental biological importance [21].

## Figures and Tables

**Figure 1 antioxidants-14-00540-f001:**
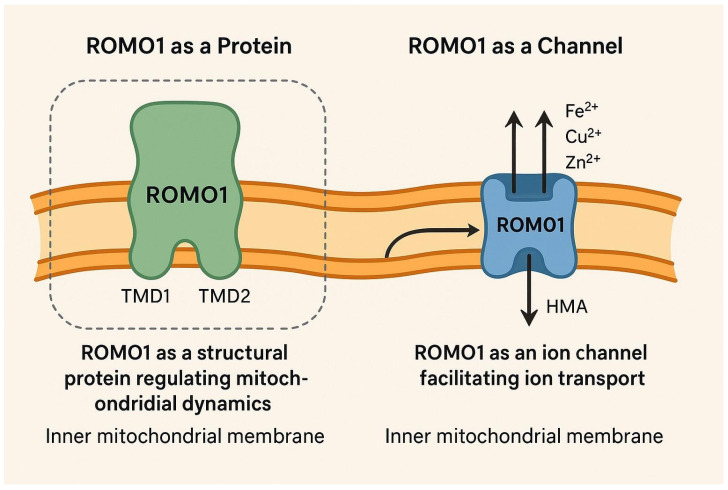
A simplified model of the inner mitochondrial membrane. On the left side, ROMO1 is illustrated as a protein with both TMD1 and TMD2 domains. On the right side, for clarification, ROMO1 is illustrated as a non-selective cation channel, where Fe/Zn/Cu ions move through (created by AI).

## Data Availability

The authors declare that all related data are available from the corresponding author upon reasonable request.

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
