# Peer review of "ROMO1: A Distinct Mitochondrial Protein with Dual Roles in Dynamics and Function"

_antioxidants, 2025, doi:10.3390/antiox14050540_

Round 1
Reviewer 1 Report
Comments and Suggestions for Authors
The subject of the presented opinion article is the characterization of the properties of the reactive oxygen species (ROS) modulator 1 (ROMO1), which belongs to mitochondrial inner membrane proteins essential for regulating mitochondrial ROS production and redox sensing. It is involved in cellular processes, such as cell proliferation, senescence, and death.
Reactive oxygen species (ROS) modulator 1 (ROMO1) is a mitochondrial membrane protein that regulates mitochondrial ROS production and redox sensing. ROMO1 regulates ROS generation within cells and is involved in cellular processes such as cell proliferation, senescence, and death.
This article presents ROMO1 from its structural and functional characteristics, physiological role, and
its implications in health and disease.
This paper broadly analyzes ROMO1 properties and functions as a ROS modulator and non-selective ion channel. This article also presents its functional properties and therapeutic potential to inhibit cancer progression. It is also a key player in ROS generation.
ROMO1 consists of two transmembrane proteins. TMD1 comprises a hydrophobic alfa helix, and TMD2 contains polar amino acids.
Its hexameric structure was determined by structural bioinformatics analysis, which proposed that ROMO1 contains two inner transmembrane proteins that interact with other mitochondrial proteins.
This article showed that ROMO1 could function as a hexameric, non-selective cation channel, similar to the function of viroporins.
The authors also presented the physiological role of ROMO1.
ROMO1 is one of the proteins generating ROS using complex III of the mitochondria electron transport chain as a signaling molecule in the cellular pathway.
It influences the concentration of ROS within the cell.
The imbalanced ROS levels can induce cellular damage, inflammation, and DNA mutations, contributing to cancer development. These reactive are responsible for over 95% of tissue damage leading to cancer.
Most endogenous reactive oxygen species (ROS) are produced in the mitochondrial respiratory chain. An imbalance in ROS production alters the intracellular redox homeostasis triggers.
This study identified a novel protein, reactive oxygen species modulator 1 (Romo1), which is localized in the mitochondria. Romo1 was found to increase the level of ROS in the cells. Increased Romo1 expression was observed in various cancer cell lines. This suggests that the increased Romo1 expression during cancer progression may cause persistent oxidative stress to tumor cells, which can increase their malignancy.
The advantage of this article could be an additional figure that will schematically present the wide activity of ROMO1 in the cells.
I have not special questions.
Author Response
Author’s Reply:
Thank you for this insightful comment.
We have revised Figure 1 to improve its clarity and informational value. The updated figure now distinctly separates ROMO1’s roles:
- On the left, ROMO1 is depicted as a structural mitochondrial protein with labeled transmembrane domains (TMD1 and TMD2), embedded in the inner membrane.
- On the right, ROMO1 is illustrated as a hexameric ion channel facilitating the transport of Fe²⁺, Zn²⁺, and Cu²⁺, with solid arrows showing ion flow.
- Dotted outlines are used to differentiate the protein’s structural role from its channel activity.
- A consistent color-coding scheme is used to visually separate the two functions, and descriptive labels have been added for clarity.
Reviewer 2 Report
Comments and Suggestions for Authors
Considering ROMO1's ability to function as a non-selective cation channel and presumably capable of transporting Fe2+, is ROMO1 affiliated with ferroptosis?
Fig 1 is unhelpful and should be reworked.
Section 3.1 is too focused on cancer.
Where is the citation for the study described on lines 151 - 160?
Recommend adding the following citations:
Sandamalika WMG, Udayantha HMV, Liyanage DS, Lim C, Kim G, Kwon H, Lee J.Fish Shellfish Immunol. 2022 Jun;125:266-275. doi: 10.1016/j.fsi.2022.05.026. Epub 2022 May 14. PMID: 35580797
Zhou D, Sun MH, Lee SH, Cui XS. Cell Div. 2021 Dec 16;16(1):7. doi: 10.1186/s13008-021-00076-7. PMID: 34915903
Shin JA, Chung JS, Cho SH, Kim HJ, Yoo YD. Biochem Biophys Res Commun. 2013 Sep 20;439(2):315-20. doi: 10.1016/j.bbrc.2013.07.012. Epub 2013 Jul 15. PMID: 23867822
Kim JJ, Lee SB, Park JK, Yoo YD. Cell Death Differ. 2010 Sep;17(9):1420-34. doi: 10.1038/cdd.2010.19. Epub 2010 Mar 5. PMID: 20203691
Lee SB, Kim JJ, Kim TW, Kim BS, Lee MS, Yoo YD. Apoptosis. 2010 Feb;15(2):204-18. doi: 10.1007/s10495-009-0411-1.PMID: 19904609
Chung JS, Lee SB, Park SH, Kang ST, Na AR, Chang TS, Kim HJ, Yoo YD. Free Radic Res. 2009 Aug;43(8):729-37. doi: 10.1080/10715760903038432. PMID: 19513905
Na AR, Chung YM, Lee SB, Park SH, Lee MS, Yoo YD. Biochem Biophys Res Commun. 2008 May 2;369(2):672-8. doi: 10.1016/j.bbrc.2008.02.098. Epub 2008 Feb 29. PMID: 18313394
Chung YM, Kim JS, Yoo YD. Biochem Biophys Res Commun. 2006 Sep 1;347(3):649-55. doi: 10.1016/j.bbrc.2006.06.140. Epub 2006 Jul 5. PMID: 16842742
Author Response
Reviewer Comment:
“Considering ROMO1’s ability to function as a non-selective cation channel and presumably capable of transporting Fe²⁺, is ROMO1 affiliated with ferroptosis?”
Author’s Reply:
Thank you for this insightful comment. We agree that ROMO1’s ability to transport Fe²⁺ raises a relevant question regarding its potential involvement in ferroptosis—a form of regulated cell death characterized by iron-dependent lipid peroxidation.
In response, we have revised the conclusion section to include a discussion on this point. While there is currently no direct experimental evidence linking ROMO1 to ferroptosis, we have acknowledged that its ion channel function and ability to modulate mitochondrial Fe²⁺ levels suggest a plausible mechanistic connection worth further investigation. We have also cited relevant literature that defines the role of iron in ferroptosis (Li W et al., 2022) to provide context.
Reviewer Comment:
“Figure 1 is unhelpful and should be reworked.”
Author’s Reply:
Thank you for your valuable feedback. We agree that the original figure did not sufficiently highlight the dual functionality of ROMO1.
In response, we have revised Figure 1 to improve its clarity and informational value. The updated figure now distinctly separates ROMO1’s roles:
- On the left, ROMO1 is depicted as a structural mitochondrial protein with labeled transmembrane domains (TMD1 and TMD2), embedded in the inner membrane.
- On the right, ROMO1 is illustrated as a hexameric ion channel facilitating the transport of Fe²⁺, Zn²⁺, and Cu²⁺, with solid arrows showing ion flow.
- Dotted outlines are used to differentiate the protein’s structural role from its channel activity.
- A consistent color-coding scheme is used to visually separate the two functions, and descriptive labels have been added for clarity.
We hope this revised figure addresses your concerns and more effectively communicates the dual role of ROMO1.
Reviewer Comment:
Where is the citation for the study described on lines 151-160
Author’s reply:
Thank you for pointing that out! Added as number 11 now on the revised manuscript.
Reviewer Comment:
Recommend adding the following citations:
Author’s reply:
Thank you for pointing this out. We appreciate the suggestion and have carefully reviewed each of the recommended references. We agree that these studies contribute valuable insights into ROMO1’s structure, function, and biological relevance.
We have now incorporated the suggested citations into the revised manuscript at appropriate locations, including the structural and functional descriptions,the pathophysiological implications sections and the conclusion. Each reference has been properly formatted and added to the reference list.
In particular:
- Sandamalika et al., 2022 is cited in the conclusion for evolutionary insights into ROMO1.
- Zhou et al., 2021 is cited in the conclusion, referencing ROMO1’s role in embryonic mitochondrial metabolism.
- Shin et al., 2013, Kim et al., 2010, and Na et al., 2006 are cited in Section 3.1 discussing ROMO1 and ROS-related cell death.
- Lee et al., 2001 and Chung et al., 2009 are cited in Conclusion section for foundational work on ROMO1 structure and its role in cell cycle regulation.
We hope the inclusion of these references strengthens the manuscript and meets your expectations.